# Ripening Process of Tomato Fruits Postharvest: Impact of Environmental Conditions on Quality and Chlorophyll *a* Fluorescence Characteristics

Hyo-Gil Choi [1,2,*] and Kyoung-Sub Park [3,*]

1 Department of Horticulture, Kongju National University, Yesan 32439, Republic of Korea
2 Agriculture and Fisheries Life Science Research Institute, Kongju National University, Yesan 32439, Republic of Korea
3 Department of Horticultural Science, Mokpo National University, Muan 58554, Republic of Korea
* Correspondence: hg1208@kongju.ac.kr (H.-G.C.); unicos75@mokpo.ac.kr (K.-S.P.)

**Abstract:** This study aimed to investigate the combined effects of temperature and light conditions on tomato maturation. Tomato fruits that had completed volumetric growth at the mature green stage were harvested and matured in growth chambers composed of two temperature conditions (daytime–nighttime: 30–20 °C or 20–15 °C) and two light conditions (0 µmol·m$^{-2}$·s$^{-1}$ or 400 µmol·m$^{-2}$·s$^{-1}$), which were set for 12 h each day and night. Our findings indicate that tomato ripening was significantly influenced by both light and temperature. Tomatoes that matured under low-temperature conditions in the absence of light took more than three times longer to transition from the green stage to the breaker stage compared to those matured under high-temperature conditions with light exposure. Notably, tomato fruit maturation occurred at a faster rate under low-temperature and light conditions than under high-temperature and dark conditions. Changes in chlorophyll *a* fluorescence parameters were observed throughout the ripening process of tomato fruits. Tomato fruits ripened under low-temperature and dark conditions exhibited significantly lower NPQ (non-photochemical quenching) and $R_{FD}$ (relative fluorescence decrease) values compared to other treatments, while their $F_O$ (initial fluorescence) and $F_M$ (maximum fluorescence) values were higher. The accumulation of sugar in tomato fruits was observed to be more influenced by light than temperature. On the other hand, the highest levels of phenolic content and lycopene were observed in tomato fruits matured under high-temperature and light conditions. Antioxidant activities, as measured by ABTS and DPPH assays, were highest in mature tomato fruits under high-temperature and light conditions, while they were lowest in fruits under low-temperature and dark conditions. In conclusion, this study highlights the critical role of temperature and light as crucial environmental factors influencing tomato maturation. Understanding these factors can contribute to optimizing postharvest conditions and enhancing fruit quality in the tomato industry.

**Keywords:** light; lycopene; maturity stage; sugar content; temperature





## 1. Introduction

Tomato (*Solanum lycopersicum* L.) is one of the most widely grown horticultural crops worldwide [1], and is used both for raw consumption and for processing. In Korea, where the four seasons are distinct, tomato is mainly cultivated in a greenhouse as an annual plant and consumed raw [2]. Various metabolites of tomatoes are known to reduce the risk of diseases such as cancer, heart disease, and cardiovascular diseases [3,4]. Lycopene is the main carotenoid found in tomatoes, accounting for about 80% of all carotenoids produced [5]. It is well-known for its high antioxidant effects [6]. The color change in tomato fruit promoted by carotenoids, such as lycopene, is one of the major ripening characteristics [7]. As tomatoes are climacteric fruits, their synthesis is closely related to the

rise in ethylene production. This process can even be triggered in detached fruit during storage periods [8].

Tomato fruits undergo significant changes in the content of sugars, organic acids, lycopene, phenolic compounds, and other phytochemicals as they ripen from the mature green stage to the red stage [9]. Furthermore, depending on the environmental conditions during the ripening process from the mature green stage, tomatoes exhibit significant differences in the accumulation of sugars, organic acids, and other components in the fruit [10]. The interaction of soluble sugars and organic acids in tomato fruits is an important quality factor due to its direct influence on the sweetness and sourness, which contribute to the overall taste intensity of tomatoes [11]. The antioxidant activity of tomato fruit is an important quality factor and is verified through analyses such as ABTS and DPPH radical quenching [12]. This antioxidant activity is influenced by conditions during the ripening process, and the ripening of tomatoes is accompanied by a gradual increase in various antioxidant compounds and a corresponding increase in antioxidant activity [13].

One of the most important environmental factors in the ripening process of tomatoes is temperature. The number of days required for tomato fruits to progress from the green stage to the breaker stage during ripening decreases as the temperature increases towards the optimum temperature for growth [14]. The content of lycopene increases during the ripening process of tomatoes, and it is know that this increase is greatly affected by temperature [15]. Ambient light is another important environmental factor affecting the color change of tomato fruit during the ripening process [1,16]. It is known that the phytochrome inherent in fruits plays a crucial role in regulating various aspects of tomato fruit ripening, including the accumulation of carotenoids such as a lycopene [17,18].

Horticultural crops, as living organisms, rely on photosynthesis to generate their own energy, making light an essential environmental factor for their growth, just like temperature. Light not only affects plant photosynthesis but also plays a vital role in signaling various metabolic processes [19,20]. One such process influenced by light conditions is the metabolism of ethylene, a hormone closely associated with plant maturation and aging [21]. In the case of postharvest peaches, exposure to blue light during ripening triggers the upregulation of ethylene biosynthesis genes, resulting in an accelerated fruit ripening process [22]. Similarly, when strawberries are exposed to light during ripening, the genes responsible for controlling pigmentation are upregulated, contributing to enhanced pigmentation [23]. Furthermore, treating harvested vegetables or fruits with LED light has been shown to have significant positive effects on quality and shelf life extension [24]. LED light treatment after harvesting can enhance the overall quality and extend the freshness of these produce items. Moreover, light has been found to impact the quality of tomato fruits by modulating the levels of soluble sugars, lycopene content, antioxidant activity, and organic acid content during the ripening process [25,26]. These findings highlight the multifaceted influence of light on the ripening of horticultural crops, showcasing its importance in horticultural crop postharvest practices.

The chlorophyll *a* fluorescence technique is a valuable tool for the non-destructive analysis of physiological responses, allowing confirmation of the plant's condition in response to various environmental conditions [27,28]. In addition to diagnosing the plant's stress state based on its cultivation environment [29], recent studies have also been conducted to classify the ripening stages of tomato fruits using chlorophyll *a* fluorescence [30,31]. In addition to representing plant conditions through the analysis of chlorophyll *a* fluorescence parameters, studies are being conducted to visually express it through images [32,33].

Therefore, this study was conducted to determine the extent to which tomato fruit is affected by temperature and light, which are the most sensitive environmental factors in the ripening process. Additionally, the study analyzed the chlorophyll fluorescence reaction to assess the differences in ripening stages based on each parameter's value and fluorescence image. The study also aims to detect fruits at the right time for harvesting through non-destructive chlorophyll *a* fluorescence imaging.

## 2. Materials and Methods

### 2.1. Plant Materials and Storage Conditions

The tomatoes (*Solanum lycopersicum* L. cv. Red 244) were grown hydroponically using a rock-wool medium in a Benno-type greenhouse located in Nonsan, Korea. The tomato fruits were harvested three times between December 2022 and March 2023 to repeat the experiment three times. On the morning of the experimental treatment, following volumetric growth, fruits in the mature green stage of the Red 244 cultivar were harvested, with an average weight of approximately 140 g. The harvested tomato fruits were transferred to growth chambers with different temperature and light conditions (Figure 1), where they were allowed to ripen fully until they reached the red stage.

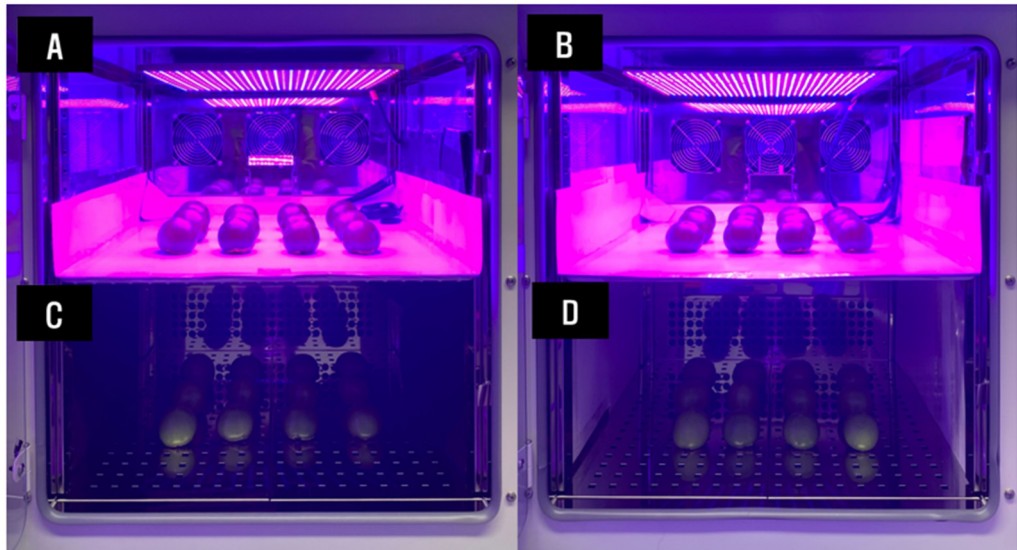

**Figure 1.** Tomato ripening under four different environmental conditions with varying temperatures and light levels. (**A**) The day and night temperatures were set at 30 °C and 20 °C, respectively, with a light intensity of 400 μmol·m$^{-2}$·s$^{-1}$ (STL). (**B**) The day and night temperatures were set at 20 °C and 15 °C, respectively, with a light intensity of 400 μmol·m$^{-2}$·s$^{-1}$ (WTL). (**C**) The day and night temperatures were set at 30 °C and 20 °C, respectively, with a light intensity of 0 μmol·m$^{-2}$·s$^{-1}$ (STD). (**D**) The day and night temperatures were set at 20 °C and 15 °C, respectively, with a light intensity of 0 μmol·m$^{-2}$·s$^{-1}$ (WTD).

In the management of tomato greenhouses in Korea, the cooling system is activated when the daytime and nighttime temperatures in summer exceed 30 °C and 20 °C, respectively. During winter, the average greenhouse temperature hovers around 20 °C, and heating is initiated when the night temperature drops below 15 °C. Furthermore, considering the performance of the artificial light used, an intensity of 400 μmol·m$^{-2}$·s$^{-1}$ is considered suitable for the efficient application of artificial light sources in greenhouses. Consequently, this study was designed to reflect these conditions. To create an environment in the growth chamber similar to that of a greenhouse, the day and night time were adjusted to 12 h each, and the relative humidity was maintained at 50–60%. As the light source for the experiment, an LED composed of blue and red in a ratio of 2:8 was used. The growth chambers with different temperature and light conditions were configured as follows:

- STL—the day and night temperatures were set at 30 °C and 20 °C, respectively, with a light intensity of 400 μmol·m$^{-2}$·s$^{-1}$;
- WTL—the day and night temperatures were set at 20 °C and 15 °C, respectively, with a light intensity of 400 μmol·m$^{-2}$·s$^{-1}$;
- STD—the day and night temperatures were set at 30 °C and 20 °C, respectively, with a light intensity of 0 μmol·m$^{-2}$·s$^{-1}$;

- WTD—the day and night temperatures were set at 20 °C and 15 °C, respectively, with a light intensity of 0 $\mu$mol·m$^{-2}$·s$^{-1}$.

## 2.2. Chlorophyll a Fluorescence

The chlorophyll *a* fluorescence induction kinetics of tomato fruits were measured after classifying the fruits into six ripening stages: mature green (where the surface of the tomato is completely green in color), breaker (with color transitioning from green to tannish-yellow, pink or red on no more than 10% of the surface), turning (where more than 10% but not more than 30% of the surface shows color ranging from green to tannish-yellow, pink, red, or a combination thereof), pink (where more than 30% but not more than 60% of the surface, in aggregate, displays pink or red coloration), light red (where more than 60% of the surface, in aggregate, shows pinkish-red or red), and red (where more than 90% of the surface, in aggregate, exhibits a red coloration). These ripening stages are defined according to the United States standards for grades of fresh tomatoes [34]. Three fruits were measured three times for each treatment. Prior to the chlorophyll *a* fluorescence analysis, the fruits were allowed to adapt to darkness for 30 min. The measurements were conducted using a fluorometer imaging system (FluorCam FC800; Photon Systems Instruments, Brno, Czech Republic) equipped with a progressive scan CCD camera (wavelength range: 400–1000 m; 720 × 560 pixels) and a prime lens (Fujinon HF8XA-1). The light panel consisted of four light-emitting diodes, including two red-orange light (620 nm) and two cool white light. The saturating super pulse light had an intensity of 4000 $\mu$mol·m$^{-2}$·s$^{-1}$, while the actinic light could reach up to 2000 $\mu$mol·m$^{-2}$·s$^{-1}$. The global light settings were set to 62% output for actinic light and 32% output for the super pulse light. The confirmation of chlorophyll *a* fluorescence images at different fruit ripening stages was performed using the protocol menu (quenching-Act1 analysis) of the FluorCam 7 software (FluorCam 7; Photon Systems Instruments, Brno, Czech Republic).

## 2.3. Analysis of Sugars, Acids, Phytochemicals, and Antioxidant Activity

To confirm the quality of phytochemicals in tomato fruits ripened under different environmental conditions of temperature and light, the red stage of the fruits was homogenized (Polytron, PT-MR3110D, Kinematica, Malters, Switzerland), and the extracts were centrifuged using a 64R Centrifuge (Beckman Coulter Inc., Brea, CA, USA) at 16,000× *g* for 30 min at 4 °C. The supernatant obtained after centrifugation was filtered through filter paper (Whatman No. 2, Sigma-Aldrich Co., St. Louis, MO, USA). The samples were stored in a cryogenic freezer at −70 °C, thawed, and then used for analysis.

After diluting 0.1 mL of the sample extracted from the fruit in 10 mL of distilled water, the sugar and acidity were measured using a fruit sugar–acidity meter (GMK-706R, G-WON Hitech Co., Ltd., Seoul, Republic of Korea). For the analysis of fructose, glucose, and sucrose in the fruit, fruit extracts were analyzed using an HPLC system (YL9100, Younglin Co., Anyang, Republic of Korea) equipped with a refractive index detector (YL9170 RI, Younglin Co., Anyang, Republic of Korea) and a Sugar-Pak column (4.6 mm × 250 mm, Supelco, Bellefonte, PA, USA) as previously described [35].

The lycopene analysis of extracts was conducted using an HPLC system (YL9100, Younglin Co., Anyang, Republic of Korea) equipped with a DB-C18 column (4.6 mm × 150 mm, Supelco, Bellefonte, PA, USA) and a DA detector (YL9120, Younglin Co., Anyang, Republic of Korea, as previously described [36].

The phenolic content in the tomato fruit extracts was determined using a UV–visible spectrophotometer (Evolution 300, Thermo Fisher Scientific., Waltham, MA, USA), with the gallic acid equivalents set as the standard. To accomplish this, aliquots of the extracts were sequentially treated with Folin–Ciocalteu reagent (50%) and $Na_2CO_3$ (20%). The resulting mixture was then incubated at 37 °C for 45 min. The absorbance of the test solutions was measured at 750 nm using the spectrophotometer, with the reagent used as a blank. The development of a blue color in each tube indicated the presence of phenolic compounds.

The antioxidant activity was assessed using methods described by [37], including the measurement of 2,2-diphenyl-1-picrylhydrazyl (DPPH) and 2,2′-azinobis-(3-ethylbenzothiazoline-6-sulfonate) (ABTS) radical scavenging abilities. The sample was mixed with the DPPH solution and allowed to react at room temperature for 30 min. The absorbance of the mixture was then measured at a wavelength of 517 nm to calculate the electron donating ability (EDA, %). EDA (%) was determined using the formula $[1 - ABS/ABC] \times 100$, where ABS represents the absorbance of the sample and ABC represents the absorbance of the control. For the ABTS assay, a mixture of 7.4 mM ABTS and 2.6 mM $K_2S_2O_8$ in a one-to-one ratio was prepared. The mixture was then filtered in the dark using Whatman No. 2 filter paper (Sigma-Aldrich Co., St. Louis, MO, USA). After overnight incubation, the mixture was adjusted to the desired absorbance at 734 nm by mixing with MeOH, and this value was used as the control. To measure the ABTS radical scavenging activity, the volume-adjusted ABTS solution and the sample were mixed together and allowed to react in a constant-temperature water bath at 37 °C for 1 min. The absorbance of the resulting mixture was then measured.

### 2.4. Experimental Design and Statistical Analysis

This experiment, aimed at confirming the changes in the ripening process of tomatoes under different environmental conditions such as temperature and light, was repeated three times using a random block design. The results of this experiment were analyzed using analysis of variance (ANOVA) with Duncan's multiple range test at a significance level of $p < 0.05$. Additionally, a two-way ANOVA was conducted using the SAS 9.4 (SAS Institute Inc., Cary, NC, USA).

## 3. Results

### 3.1. Changes of Color and Time to Maturity of Tomato Fruits

The ripening stages of tomatoes are generally classified into six stages: mature green, breaker, turning, pink, light red, and red. Figure 2A illustrates the color changes observed during these ripening stages in the growth chamber under different environmental conditions in this experiment. Tomatoes ripened in STL and WTL, which are subjected to light treatment, exhibited distinct difference across all ripening stages. However, in the cases of STD and WTD, which were not exposed to light irradiation, the progression through the six ripening stages was not clearly discernible. Notably, tomatoes ripened without light irradiation did not change to red as a whole, and the fruit surface displayed partial discoloration.

Figure 2B depicts the number of days required for tomato fruits to ripen under four distinct environmental conditions, featuring varying temperatures and light levels. Among the six stages of tomato ripening, a significant difference was observed in the number of days required to transition from the mature green stage to the breaker stage due to difference environmental conditions. However, during the ripening process of tomato fruit from the breaker stage to the red stage, there was no significant difference observed in the aging period based on environmental conditions. The results of the two-way ANOVA analysis reveal that both temperature and light significantly influence the ripening of tomato fruits. When comparing STL, which was subjected to high temperature and light irradiation, with WTD, which did not receive light irradiation and low temperature, the duration required for ripening was 2.5 times longer in WTD.

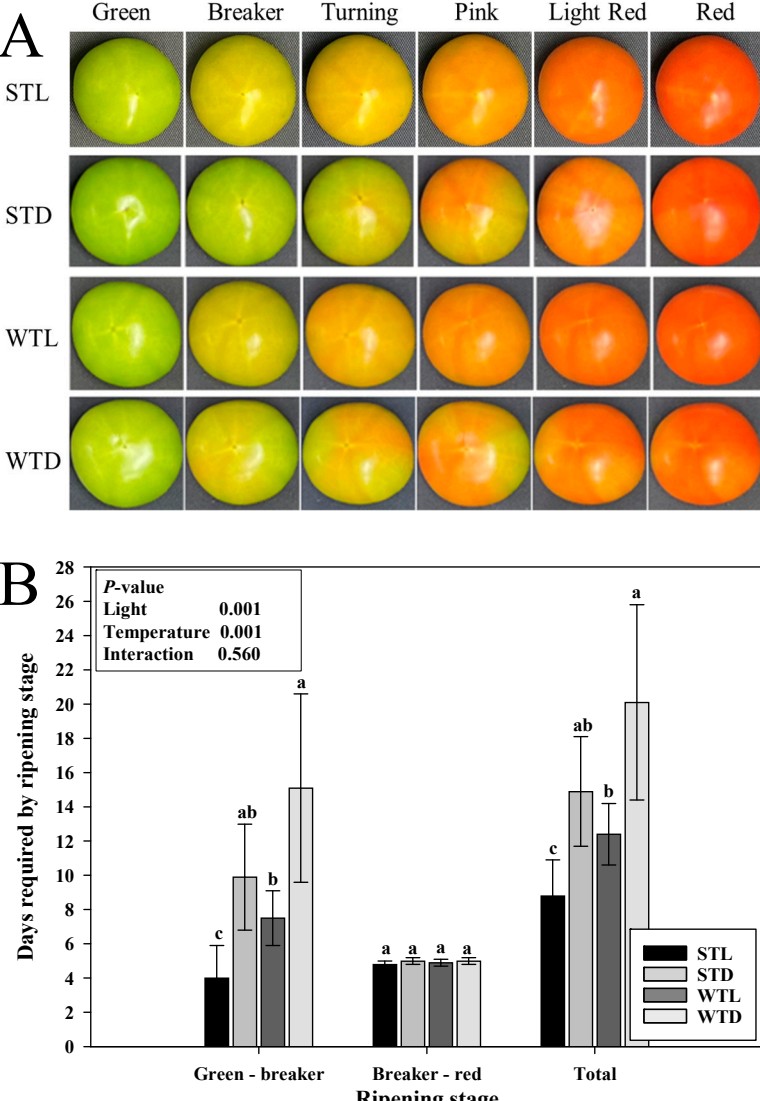

**Figure 2.** Changes of color (**A**) and time (**B**) to maturity of tomato fruit at different ripening stages under four distinct environmental conditions with varying temperatures and light levels. Vertical bars represent standard deviations (*n* = 30). Small letters at the data points indicate mean separation between the values using DMRT (*p* = 0.05). *p* values were determined through two-way ANOVA.

### 3.2. Chlorophyll a Fluorescence in Tomato Fruits

Upon examining the parameter-specific image of chlorophyll *a* fluorescence based on the ripening stages of tomato fruits, distinct differences in fluorescence were observed among the various ripening stages (Figure 3). Notably, consistent patterns of fluorescence images were observed throughout the measurements of various fruit samples at different ripening stages. In the fluorescence image analysis of tomato fruit ripening stages, no significant difference was observed in the minimum chlorophyll fluorescence in the dark-adapted state ($F_O$) or the maximum chlorophyll fluorescence in the dark-adapted state ($F_M$) values, assessed by ripening stage. However, the maximum PSII quantum yield ($Q_Y$) images showed distinct groupings across four stages: mature green stage, breaker stage, turning and pink stages, and light red and red stages. Similarly, the fluorescence decline ration in light ($R_{FD}$) images also exhibited distinct groupings across the mature green stage, the breaker and turning stages, and the pink stage, as well as the light red and red stages. Furthermore, it can be observed in the non-photochemical quenching (NPQ) image that significant changes in fluorescence occurred throughout the ripening stages. In particular,

the transition from the mature green stage to the breaker stage, which is significantly influenced by environmental factors during the ripening process of tomatoes, could be distinctly distinguished using the chlorophyll *a* fluorescence image of the fruit.

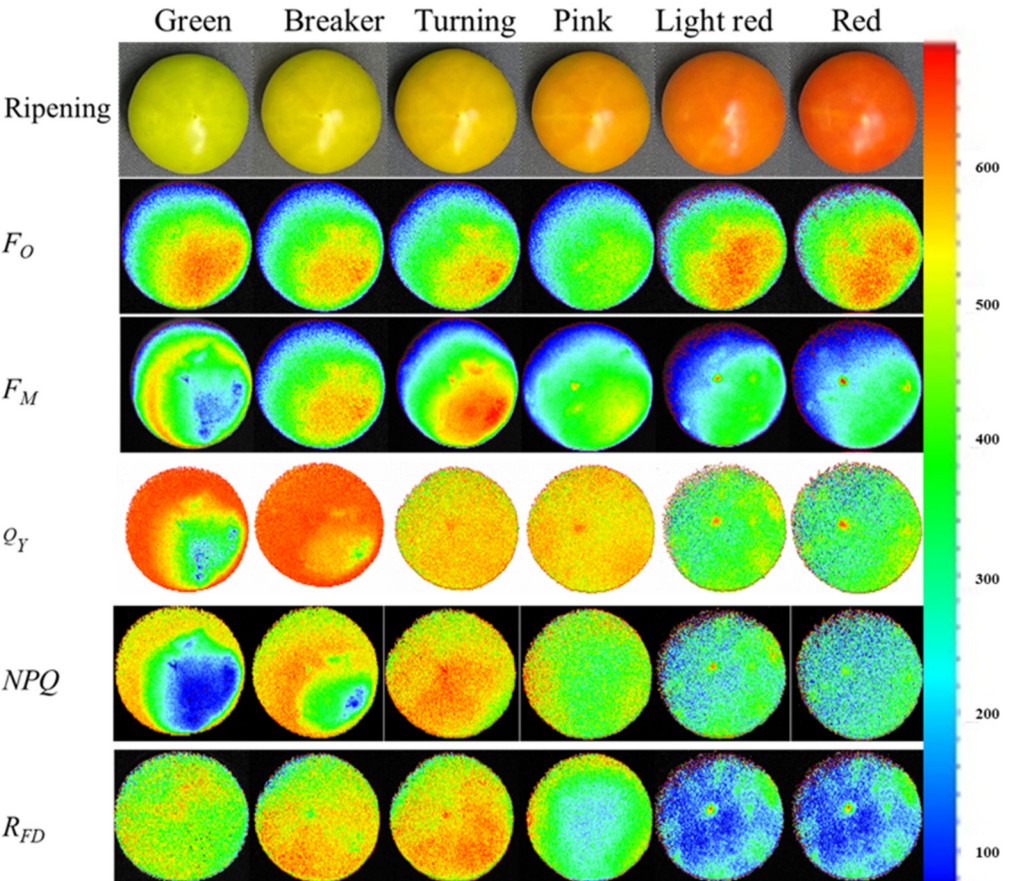

**Figure 3.** Fluorescence images of tomato fruit based on parameter values of chlorophyll *a* fluorescence under four different environmental conditions with varying temperatures and light levels. $F_O$: the minimum chlorophyll fluorescence in dark-adapted state; $F_M$: the maximum chlorophyll fluorescence in dark-adapted state; $Q_Y$: the maximum PSII quantum yield; NPQ: non-photochemical quenching; $R_{FD}$: the fluorescence decline ration in light.

Figure 4 shows the variations in chlorophyll *a* fluorescence parameter values at each stage of ripening for tomato fruits ripened in growth chambers under different environmental conditions. The $F_O$ values in tomato fruits ripened under low-temperature conditions (WTL and WTD) exhibited an increasing trend until the pink stage, followed by a decrease. In contrast, fruits ripened under high temperature conditions (STL and STD) demonstrated a consistently decreasing pattern as ripening progressed. While the chlorophyll *a* fluorescence parameter values (except the $F_O$ parameter) for each ripening stage of tomato fruits ripened under WTD conditions showed a gradual increase or decrease, the fluorescence parameter values of tomatoes ripened under STL conditions exhibited rapid increase or decrease. This indicates that the physiological changes in tomato fruits can be significantly determined by environmental conditions.

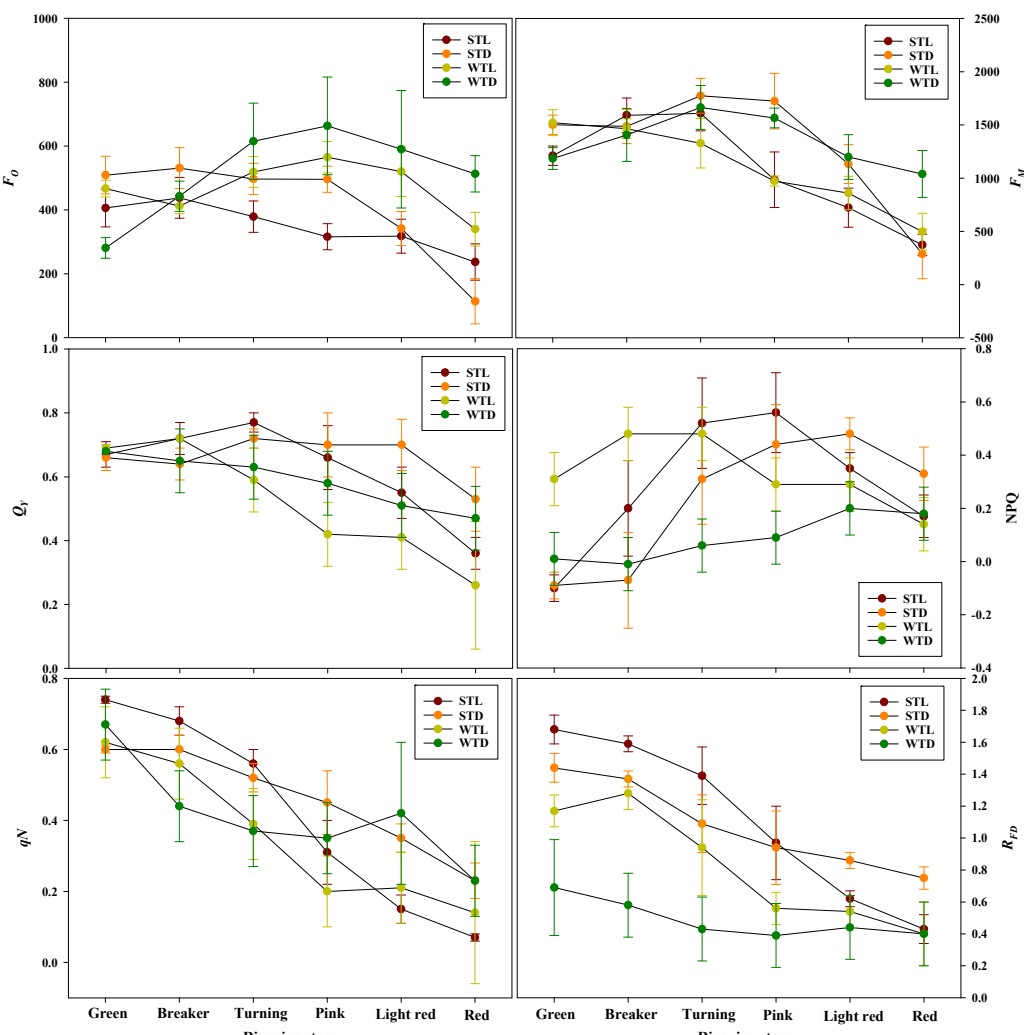

**Figure 4.** Changes in chlorophyll *a* fluorescence parameter values during the ripening of tomato fruit under four different environmental conditions with varying temperatures and light levels. Vertical bars are standard deviations ($n = 9$). $F_O$: the minimum chlorophyll fluorescence in the dark-adapted state; $F_M$: the maximum chlorophyll fluorescence in the dark-adapted state; $Q_Y$: the maximum PSII quantum yield; NPQ: non-photochemical quenching; $q_N$: non-photochemical quenching of variable fluorescence; $R_{FD}$: the fluorescence decline ration in light.

### 3.3. Phytochemicals and Antioxidant Activity in Tomato Fruits

The effects of temperature and light conditions during the ripening process of tomato fruits on the accumulation of various phytochemicals in the fruit were analyzed (Figure 5). The accumulation of soluble sugar content in tomato fruits was significantly influenced by light, with the highest accumulation observed when exposed to light under low temperature conditions. On the other hand, in terms of the acidity of tomato fruits, those ripened under high-temperature and -light conditions (STL) or low-temperature and dark conditions (WTD) exhibited significantly higher acidity levels. The phenol content in tomato fruits was found to increase under the STL condition, while no significant difference was observed in the other treatments. The lycopene content was highest in tomato fruits ripened under STL conditions, and lowest under WTD conditions. Furthermore, there was no significant difference in lycopene content between tomato fruits ripened under STD and WTL conditions.

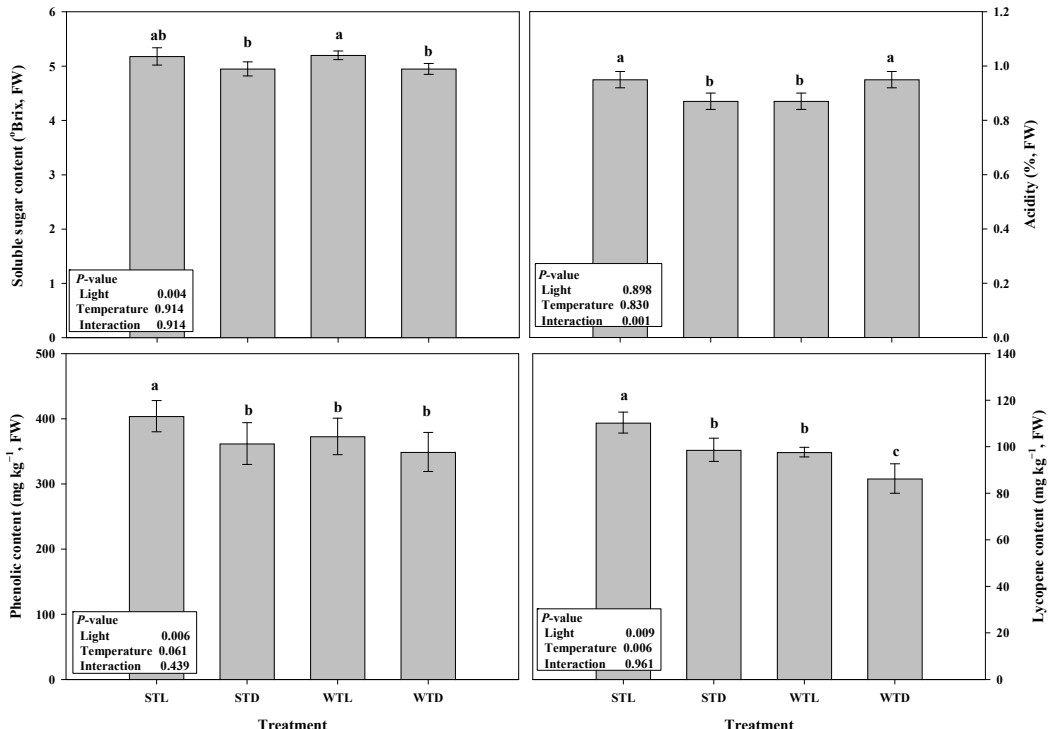

**Figure 5.** Phytochemical content of red stage matured tomato fruits under four different environmental conditions with varying temperatures and light levels. Vertical bars are standard deviations (*n* = 3). Small letters at the data points indicate mean separation between the values by DMRT (*p* = 0.05). *p* values were determined by two-way ANOVA.

Table 1 shows the variations in fructose, glucose, and sucrose levels in tomato fruits ripened under different environmental conditions. Irrespective of the treatment, fructose and glucose were identified as the primary sugars in tomato fruits. The content of sucrose, which is a glycoside formed by glucose and fructose, was relatively low. No significant effect of light on the sucrose content of tomato fruit was observed, whereas a substantial effect of light on fructose and glucose content was evident. Furthermore, within the temperature range examined in this experiment, no significant effect of temperature on the sugar content was observed.

**Table 1.** Soluble sugars content of red stage matured tomato fruits under four different environmental conditions with varying temperatures and light levels.

| Treatment | Fructose | Glucose | Sucrose |
|---|---|---|---|
| | mg·g$^{-1}$ FW | | |
| STL | 24.49 ± 1.31 a [z] | 21.36 ± 1.11 ab | 1.19 ± 0.26 a |
| STD | 20.59 ± 1.12 b | 19.25 ± 1.45 bc | 0.97 ± 0.29 a |
| WTL | 26.71 ± 1.34 a | 22.78 ± 1.34 a | 1.25 ± 0.11 a |
| WTD | 19.87 ± 1.23 b | 17.89 ± 1.36 c | 0.98 ± 0.08 a |
| Effect (*p* value) * | | | |
| Temperature | 0.4212 | 0.9217 | 0.8417 |
| Light | 0.0003 | 0.0054 | 0.1283 |
| Interaction | 0.1371 | 0.1566 | 0.8765 |

* *p* values were determined by two-way ANOVA. [z] Average values and standard deviation are presented, and values followed by different lower case letters within a column are different (DMRT at *p* ≤ 0.05, *n* = 3).

The antioxidant activity of tomato fruits was confirmed by ABTS and DPPH assays when they were subjected to ripening under different environmental conditions of temperature and light (Figure 6). The antioxidant activity of tomato fruits varied based on

the ripening environmental conditions. Particularly, they exhibited high activity under high-temperature and -light irradiation conditions (STL), while the activity significantly decreased under low-temperature and dark conditions. When considering the ABTS activity, the impact of light was found to be significant, while there was no significant difference in the effect of temperature. However, in the case of DPPH, both light and temperature had a significant influence on the activity.

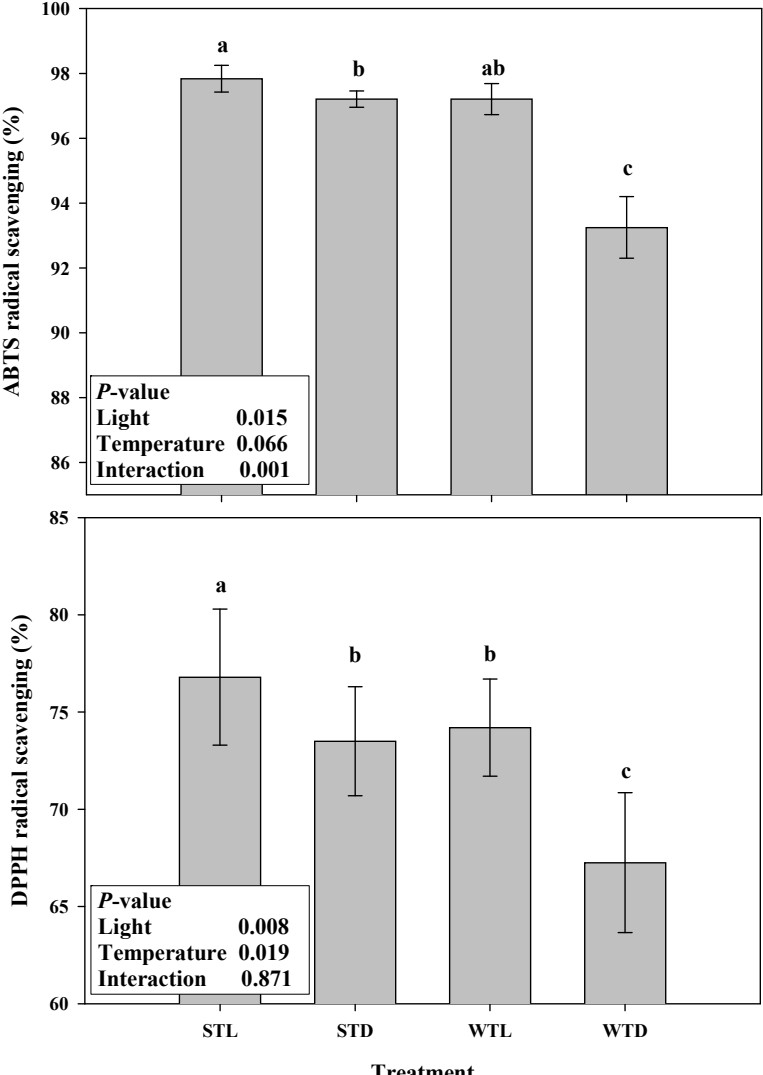

**Figure 6.** Antioxidant activity shown as ABTS and DPPH of red stage matured tomato fruits under four different environmental conditions with varying temperatures and light levels. Vertical bars are standard deviations (*n* = 3). Small letters at the data points indicate mean separation between the values by DMRT (*p* = 0.05). *p* values were determined by two-way ANOVA.

## 4. Discussion

Various environmental conditions, such as light, temperature, humidity, and $CO_2$, play an important role in the pre-harvest and post-harvest maturation of horticultural crops [38–40]. Tomato fruit, which is grown and consumed in various countries around the world, exhibits climacteric fruit characteristics and, like other horticultural crops, is influenced by a range of environmental factors [41]. In this study, ripening tomato fruits were subjected to different temperature and light conditions. Notably, tomatoes ripened in darkness without exposure to light exhibited a visible characteristic wherein the fruit's surface displayed mottled colors of red and green (Figure 2A). It is known that tomato

pericarp cells regulate tomato carotenoid biosynthesis through light sensing [16]. Therefore, in this study, it was observed that the biosynthesis process does not occur smoothly when tomatoes mature in darkness, resulting in a mottled appearance of the tomato pericarp, indicating disrupted pigmentation patterns.

Tomatoes take the most time to transition from the mature green stage to the breaker stage compared to other ripening stages [14]. In particular, it is known that as the storage temperature increases up to 30 °C, the rate at which tomatoes transition from the mature green stage to the breaker stage is accelerated [14]. Tomatoes are primarily classified into ripening stages based on changes in skin color, and light plays a significant role in these color transformations [42]. It has been reported that the lycopene concentrations in tomato fruits exposed to LED light were higher compared to those exposed to dark conditions [42–44]. Consistent with the previous studies mentioned [14,42–44], the findings of this study demonstrate a pronounced influence of temperature and light on the ripening of tomato fruits (Figure 2). The time required for tomato fruits to transition from the green stage to the breaker stage varied up to threefold, depending on the environmental conditions during postharvest ripening. However, the period required for maturation from the breaker stage to the red stage of tomato fruit did not show a significant difference based on the treatment (Figure 2B). Based on these results, it is believed that the period when tomato fruit is most influenced by the environment is between the green stage and just before it reaches the breaker stage. Therefore, the environmental conditions during the transition period from the mature green stage to the breaker stage will greatly affect tomato ripening.

In a study utilizing spectrophotometric absorbance differences, the relative concentrations of PSII and PSI reaction centers were determined in the green fruit tissue of tomatoes [45]. According to the research report, the values of chlorophyll fluorescence parameters in tomato green fruits were found to be comparable to those measured in leaves [46]. Additionally, the RuBPCO activity, which was similar to that observed in the leaves during the immature stage of the fruit, decreased as the fruit ripened [46]. This finding suggests that chlorophyll fluorescence images can be effectively analyzed during tomato fruit ripening, given the demonstrated responsiveness of tomato fruit tissues to light. Recently, numerous studies have been conducted to utilize chlorophyll fluorescence induction for the classification of ripening stages in tomato fruits [28,47–49]. It has been reported that the non-photochemical quenching of photosystem (NPQ), one of the chlorophyll fluorescence parameters, is an effective method to classify tomato fruits according to their ripening stage [31]. The fluorescence decline ratio in light ($R_{FD}$), which is one of the chlorophyll fluorescence parameters, has also been reported as a reliable tool for classifying the ripening stage of tomatoes [47]. In this study, tomatoes were successfully classified by their ripening stage using fluorescence images obtained from chlorophyll fluorescence induction (Figure 3). Specifically, $R_{FD}$, NPQ, and $Q_Y$ images were identified as highly effective tools for ripening stage classification. In the case of $R_{FD}$ parameters, tomato fruit ripening in a location with favorable light and temperature conditions showed a steep decline as the ripening stage progressed, while it was observed to decline more gradually under unfavorable ripening conditions (Figure 4). Therefore, even without an image, the chlorophyll fluorescence induction parameter values such as $R_{FD}$ can be used as essential tools to non-destructively assess the efficiency of tomato fruit maturation during the ripening process.

Tomatoes are recognized as an important horticultural crop due to their rich content of various phytochemicals [3,4]. It is known that environmental conditions, such as light and temperature, influence the metabolites of phytochemicals in the fruit during the tomato ripening process [44]. The formation of carotenoids in climacteric fruits does not require induction by light, but shaded fruits have a lower carotenoid content [50]. Furthermore, it is reported that red light affects chlorophyll degradation, whereas carotenoid synthesis is enhanced by blue light [50]. Also, the sugar content of horticultural crops varies depending on the light conditions. Tomatoes grown in shaded areas have lower sugar contents in their

fruits compared to tomatoes grown in sunlight [1]. Similarly, strawberries grown in shaded areas, despite receiving some sunlight, exhibit reduced sugar contents compared to those grown under direct sunlight [51]. Consistent with previous studies [1,49], the present study found a significant increase in the sugar content of tomato fruits when they were irradiated with light during the ripening process, compared to those ripened in darkness (Table 1 and Figure 5). Ultimately, the biosynthesis of phytochemicals in tomatoes is regulated by the interaction between temperature and light [52].

As a result, the interaction between temperature and light is crucial in the ripening process of tomato fruits. Therefore, it is believed that by implementing LED lighting in close proximity to the fruit during the tomato cultivation process, it will be possible to enhance the quality of tomato fruits while promoting their ripening. This approach is expected to be effective not only during the winter season but also in the summer season.

**5. Conclusions**

Based on the results of this study, three key conclusions are presented. Firstly, when light is applied during the ripening stage of tomato fruits, it leads to uniform color changes and accelerates the ripening process. Secondly, chlorophyll fluorescence images and parameter values can be valuable tools for effectively classifying the ripening stages of tomato fruits. Lastly, employing LED illumination near the fruit proves to be an efficient cultivation method for growing tomatoes in both low- and high-temperature seasons.

**Author Contributions:** H.-G.C. contributed to the experimental design, conceptualization, data analysis, formal analysis, investigation, data curation, and original draft and writing; K.-S.P. contributed to the data analysis and writing. All authors have read and agreed to the published version of the manuscript.

**Funding:** This study received financial support from the Korea Institute of Planning and Evaluation for Technology in Food, Agriculture, and Forestry (IPET), Korea Smart Farm R&D Foundation (Project No. 421001-03) and Kongju National University (Project No. 2023-0223-01).

**Data Availability Statement:** The data presented in this study are available in the article.

**Acknowledgments:** We would like to express our thanks to the anonymous reviewers for their useful comments.

**Conflicts of Interest:** The authors declare no conflict of interest.

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
