# Peer review of "Ripening Process of Tomato Fruits Postharvest: Impact of Environmental Conditions on Quality and Chlorophyll a Fluorescence Characteristics"

_horticulturae, doi:10.3390/horticulturae9070812_

Round 1

Reviewer 1 Report

This study investigates the effects of light and temperature on tomato maturation and finds that low temperature and light conditions lead to faster maturation. The study also shows that light application during ripening can lead to uniform color changes and accelerate the process. Chlorophyll fluorescence imaging and parameter values are found to be useful tools for classifying tomato fruit ripening stages. The study suggests that employing LED illumination near the fruit can be an efficient cultivation method in both low and high-temperature seasons. Overall, the study provides insights into tomato maturation and suggests potential techniques for improving cultivation practices.

The following suggested improvements should be made before recommending the paper for publication:

1.      Did you compare tomato fruits' quality and chlorophyll fluorescence characteristics as they ripened under various environmental conditions with those ripened hydroponically using a rock-wool medium in a Benno-type greenhouse?

2.      Based on the study's findings, what do you think would happen if tomato fruits were ripened under temperatures higher than 30°C and lighting higher than 400 µmol·m⁻²·s⁻¹?

Author Response

We are grateful for your helpful and constructive feedback.

We have addressed comment and revised the manuscript accordingly to improve the quality of our paper. 

The modified parts of this manuscript are indicated in red.

Reviewer's point 1: Did you compare tomato fruit’ quality and chlorophyll fluorescence characteristics as they ripened under various environmental conditions with those ripened hydroponically using a rock-wool medium in a Benno-type greenhouse?

Response 1: Unfortunately, we were unable to conduct a comparative analysis between tomato fruits ripening under normal greenhouse conditions and tomato fruits ripening in a growth chamber in this experiment. Therefore, the title has been modified to focus on the postharvest ripening process, as it is directly associated with the environmental conditions that impact tomato fruit ripening after harvest. Based on the specific feedback provided by the reviewer, we have revised the title to: "Ripening Process of Tomato Fruits Postharvest: Impact of Environmental Conditions on Quality and Chlorophyll a Fluorescence Characteristics.”

Reviewer's point 2: Based on the study’s findings, what do you think would happen if tomato fruits were ripened under temperatures higher than 30°C and lighting higher than 400 μmol·m–2·s–1?

Response 2: We added the reason that information regarding the rationale behind setting the specific light and temperature conditions was included in the Materials and Methods section to aid in understanding (Lines 101-107).

Reviewer 2 Report

The authors presented a very interesting paper about the changes in the quality and chlorophyll a fluorescence characteristics of tomato fruits under various environmental conditions during ripening.

My general comment derives from the title itself – 'during ripening'. The presented research was performed only on post-harvest ripening, and we can't be sure that the same findings will apply to pre-harvest, or let's say 'regular' ripening. Therefore, the whole paper should be revised in order to underline that the research was focused on post-harvest ripening.

Also, please revise some minor comments:

1)      Lines 16-17 – The sentence was repeated.

2)      Lines 21-22 – Why is the mature green-breaker underlined? Maybe it would be more important the period mature green-red?

3)      Line 27 – the authors stated that temperature didn't affect the researched parameter, but an interaction effect was noted so please make it more clear.

4)      Lines 33-35 – Rephrase to a more consistent conclusion. The natural environment cannot be managed, only a controlled environment can.

5)      Line 149 – Do not start the sentence with a number.

The whole paper is well written and I believe that after a major revision, it can be accepted.

 Moderate editing of English language is required.

Author Response

We are grateful for your helpful and constructive feedback.

We have addressed comment and revised the manuscript accordingly to improve the quality of our paper.

Necessary corrections have been made to the text of the manuscript, incl.: we have supplemented the manuscript with introduction, methods and materials, discussion, and added references.

The modified parts of this manuscript are indicated in red.

Reviewer's point 1: My general comment derives from the title itself- ‘during ripening’. The presented research was performed only on post-harvest ripening, and we can’t be sure that the same findings will apply pre-harvest, or let’s say ‘regular’ ripening. The whole paper should be revised in order to underline that the research was focused on post-harvest ripening.

Response 1: Based on the specific feedback provided by the reviewer, we have revised the title to: "Ripening Process of Tomato Fruits Postharvest: Impact of Environmental Conditions on Quality and Chlorophyll a Fluorescence Characteristics.”

Reviewer's point 2: Lines 16-17- The sentence was repeated.

Response 2: Thank you for carefully reviewing the manuscript. The repeated sentences have been removed.

Reviewer's point 3: Lines 21-22- Why is the mature green-breaker underlined? Maybe it would be more important the period mature green-red?

Response 3: In the abstract, lines 21-22 were omitted. However, based on the reviewer's feedback, the additional explanation has been provided in the discussion section (Lines 337-345).

Reviewer's point 4: Lines 27- the authors stated that temperature didn’t affect the researched parameter, but an interaction effect was noted so pleased make it more clear.

Response 4: We corrected the sentence as follows: The accumulation of sugar in tomato fruits was observed to be more influenced by light rather than temperature (Lines 24-26).

Reviewer's point 5: Lines 33-35- Rephrase to a more consistent conclusion. The natural environment cannot be managed, only a controlled environment can.

Response 5: We have not only revised the conclusions of the abstract (Lines 30-32) but also made overall improvements to enhance tis clarity and coherence.

Reviewer's point 6: Lines 149- Do not start that sentence with a number.

Response 6: We corrected the sentence as follows: The antioxidant activity was assessed using methods described by [34], including the measurement of 2,2-diphenyl-1-picrylhydrazyl (DPPH) and 2,2’-azinobis-(3-ethylbenzothiazoline-6-sulfonate) (ABTS) radical scavenging abilities (lines 179-181).

Reviewer 3 Report

The main problem is that why did the authors set the light intensity at 400 µmol·m–2·s–1. What is the basis for temperature setting? In addition, the treatment under natural light may be set as control. Other issues need to be considered,

1. There is a lack of literature on the impacts of light on fruit ripening in the introduction section.

2. Parameter settings are missing in the fluorometer (FluorCam FC800; Photon Systems Instruments, Drasov, Czech Republic).

3. How to accurately determine the different ripening stages of tomatoes? Because it involves the data in Figure 2.

4. Will the color of tomatoes affect the results in Figure 3?

5. Literature on the fluorescence characteristics of light on fruit ripening needs to be added in the Discussion section.

6. The standard deviation is a bit large in Figure 2.

7. When to pick green fruits and what are the standards for picking them.

Author Response

We are grateful for your helpful and constructive feedback.

We have addressed comment and revised the manuscript accordingly to improve the quality of our paper.

Necessary corrections have been made to the text of the manuscript, incl.: we have supplemented the manuscript with introduction, methods and materials, discussion, and added references.

The modified parts of this manuscript are indicated in red.

Reviewer's point 1: The main problem is that why did the authors set the light intensity at 400 μmol·m–2·s–1. What is the basis for temperature setting? In addition, the treatment under natural light may be set as control.    

Response 1: We added the reason that information regarding the rationale behind setting the specific light and temperature conditions was included in the Materials and Methods section to aid in understanding (Lines 101-107).

Reviewer's point 2: There is a lack of literature on the impacts of light on fruit ripening in the introduction section.

Response 2: We corrected the introduction part by according to reviewers' comments. Information on the effect of light on tomato ripening and fruit quality has been added to the introduction section (Lines 68-77).

Reviewer's point 3: Parameter settings are missing in the fluorimeter (FluorCam FC800; Photon Systems Instruments, Drasov, Czech Republic).

Response 3: We added the parameter settings in the fluorimeter by according to reviewers' comments (Lines 142-152).

Reviewer's point 4: How to accurately determine the different ripening stages of tomatoes? Because it involves the data in Figure 2.

Response 4: We included reference to a classification for determining the ripening stage of tomatoes according to the United States standard for grading fresh tomatoes (Lines 130-140).

Reviewer's point 5: Will the color of tomatoes affect the results in Figure 3?

Response 5: We added the sentence as “Notably, consistent patterns of fluorescence images were observed throughout the measurements of various fruit samples at different ripening stages.” (Lines 231-233). 

Reviewer's point 6: Literature on the fluorescence characteristics of light on fruit ripening needs to be added in the discussion section.

Response 6: We added that literature on the fluorescence characteristics of light on fruit ripening in the discussion section (Lines 346-353).

Reviewer's point 7: The standard deviation is a bit large in Figure 2.

Response 7: Unfortunately among the ripening stages of tomatoes, the green stage requires the longest ripening time, and during this period, it was challenging to harvest samples that were visually identical. It is presumed that the large standard deviation is due to the variation between the initial green stage and the final green stage. In contrast, as the fruits mature, it becomes evident that there is minimal difference in standard deviation from the breaker stage to the red stage.

Reviewer's point 8: When to pick green fruits and what are the standards for picking them.

Response 8: We collected samples by harvesting fruits in the mature green stage of the Red 244 cultivar on the morning of the experimental treatment, after volumetric growth, with an average weight of approximately 140 g. (Lines 96-98).

Round 2

Reviewer 2 Report

The authors have satisfactorily responded to all my comments, and therefore I propose that the paper be accepted in its current form.

The authors have satisfactorily responded to all my comments, and therefore I propose that the paper be accepted in its current form.

Author Response

Thank you once again for dedicating your time to reviewing this manuscript.

Reviewer 3 Report

the revised manuscript was improved in the Introduction and M & M section. Morever, some literature on the effects of light on fruit ripening or post-harvest of fruits was suggested to be added in the Introduction (Line 68-77).

Author Response

Thanks again for your helpful and constructive feedback.

We have enhanced the introduction section based on the comment suggesting the inclusion of relevant literature on the effects of light on fruit ripening and postharvest processes.

The modified contents are denoted or indicated in red.

Comments and Suggestions for Authors: the revised manuscript was improved in the Introduction and M & M section. Morever, some literature on the effects of light on fruit ripening or post-harvest of fruits was suggested to be added in the Introduction (Line 68-77).

Author's Notes to Reviewer: We have modified the introduction by incorporating the effects of light on fruits during both postharvest and ripening processes with literature as follows: Horticultural crops, as living organisms, rely on photosynthesis to generate their own energy, making light an essential environmental factor for their growth, just like tempera-ture. Light not only affects plant photosynthesis but also plays a vital role in signaling various metabolic processes [19, 20]. One such process influenced by light conditions is the metabolism of ethylene, a hormone closely associated with plant maturation and ag-ing [21]. In the case of postharvest peaches, exposure to blue light during ripening triggers the upregulation of ethylene biosynthesis genes, resulting in an accelerated fruit ripening process [22]. Similarly, when strawberries are exposed to light during ripening, genes re-sponsible for controlling pigmentation are upregulated, contributing to enhanced pig-mentation [23]. Furthermore, treating harvested vegetables or fruits with LED light has been shown to have significant positive effects on quality and shelf life extension [24]. LED light treatment after harvesting can enhance the overall quality and extend the freshness of these produce items. Moreover, light has been found to impact the quality of tomato fruits by modulating the levels of soluble sugars, lycopene content, antioxidant activity, and organic acid content during the ripening process [25, 26]. These findings highlight the multifaceted influence of light on the ripening of horticultural crops, showcasing its im-portance in horticultural crop postharvest practices (lines 68-84).

Thank you once again for dedicating your time to reviewing this manuscript.